# Genotypic distribution and molecular epidemiology of HPV in women in the UAE using PNA-based RT PCR

Devapriya Finney[1☺*], Ahmed Luay Osman Hashim[1], Sunil Kumar Bylappa[2], Ameena Ebrahim Mugar[2], Manivannan Nandhagopal[3], Mohammed Abdulrazzaq Jabbar [4], Nazeerullah Rahamathullah [5,6☺*]

1 Department of Medical Laboratory Sciences, College of Health Sciences, Gulf Medical University, Ajman, United Arab Emirates, 2 Thumbay Laboratories, Gulf Medical University, Ajman, United Arab Emirates, 3 Department of Microbiology, Saveetha Medical College and Hospital, Saveetha Institute of Medical and Technical Sciences, Saveetha University, India, 4 Department of Community Medicine, College of Medicine, Gulf Medical University, Ajman, United Arab Emirates, 5 Department of Biomedical Sciences, College of Medicine, Gulf Medical University, Ajman, United Arab Emirates, 6 Thumbay Research Institute for Precision Medicine, Gulf Medical University, Ajman, United Arab Emirates

☺ These authors contributed equally to this work.
* devapriya@gmu.ac.ae (DP); dr.nazeer@gmu.ac.ae (NR)

## Abstract

### Objectives

This study aimed to determine the genotypic distribution and molecular epidemiology of HPVs in PAP smear samples of women in the UAE using a peptide nucleic acid (PNA)-based fluorescence melting curve analysis method.

### Methods

A cross-sectional retrospective study was conducted between January 2024 to January 2025 after receiving ethical approval from the Institutional Review Board of Gulf Medical University. A total of 229 liquid-based cervical cytology samples were obtained from women aged 20–55 years attending the Gynaecology out-patient department of Thumbay University Hospital and other hospitals of the UAE, processed for the routine cytological examination to identify and differentiate morphological changes of the PAP smears. HPV genotyping was performed using PNA-based fluorescence melting curve analysis by the RT-PCR method.

### Results

A total of 191 HPV genotypes were detected in 96 HPV positive PAP smear samples, including 47 abnormal cytology and 49 NILM samples. 137 HR and 54 LR-HPV genotypes were identified in all 96 positive samples. The highest rate of mixed HR and LR-HPV genotypes (18%) was detected in women aged 31–40 years old. HR53, 16, 68, 66, 31, 35, and LR6 were the predominant genotypes. Mixed HR and LR-HPV

**Data availability statement:** All the relevant data are within the manuscript and its Supporting information files & Supplementary tables 1, 2, 3 & 4.

**Funding:** The author(s) received no specific funding for this work.

**Competing interests:** The authors have declared that no competing interests exist.

infection is present in 26% cases; in that HPV53 was the primary genotype, followed by HPV35, 66, 11, 43, 81, 61, and 6.

## Conclusion

The genotypes 53, 16, 68, 66, 31, 61, 35, and 6 were the most common genotypes detected in the PAP smear samples. Notably, 21% of normal epithelial cells of the PAP smear samples tested positive for different HR-HPV genotypes. These findings underscore the crucial role of HPV genotyping using RT-PCR, enhancing the effectiveness of screening surveillance and providing foundational data for future prospective studies, vaccination impact assessments, and targeted screening strategies.

## Introduction

Globally, cervical cancer, primarily caused by major oncogenic Human Papillomavirus (HPV) genotypes, is the 3rd most common cancer in women, predominantly affecting middle-aged women. According to the World Health Organization (WHO), globally, cervical cancer is the 4th most common cancer in women, with around 660,000 new cases in 2022. Furthermore, in 2022, about 94% of the 350,000 deaths caused by cervical cancer occurred in low and middle-income countries. In the UAE, it is the 5th most common cause of cancer and the 3rd most frequent cancer among women between 15–44 years of age. It is the 13th leading cause of cancer-related mortality in the UAE. HPVs are small, double-stranded DNA viruses transmitted through sexual mode [1]. Most people who are sexually active encounter HPV infection at some point in their lives. To date, over 450 HPV genotypes have been identified, with approximately 40 capable of infecting the anogenital tract. While many are associated with cervical cancer, others are rarely or never found in large cancer studies, leading to their classification as 'high-risk' (HR) or 'low risk' (LR) [2–5].

A persistent infection with HR HPVs is associated with several human carcinomas, especially cervical cancer [5]. Women are at high risk of cervical infection, as evidenced by the PAP smear [6,7]. Not all individuals develop genital warts or progress to malignancy; some HR HPV infections go unnoticed without symptoms and are treated due to the individual's healthy immune status. Sometimes, HR HPV infection escapes the immune system and can lead to changes in the squamous epithelia, resulting in squamous cell carcinoma [8]. HR HPV genotypes 16, 18, 31, 33, 35, 39, 45, 51, 52, 56, 58, 59, and 66 have been associated with cervical carcinogenesis, as well as with anogenital, head, and neck cancers [9,10]. LR genotypes 6 and 11 are regularly detected in benign or low-grade cervical tissue morphology changes. 70% of cervical cancer cases are mainly caused by HR genotypes 16 and 18 [11,12].

This is why cervical screening has been introduced for primary screening to detect cancer. According to the 2023 updates from the WHO and the National Cancer Institute, HPV-related cervical cancer has become a global health burden. In the United States, 90% of anorectal cancer, over 60% of penile cancers, 75% of vaginal cancer, and 70% of vulvar cancer are due to HPV. Routine screening can prevent the

progression to the malignant stage by removing precancerous cells before they develop into cancer [1,6]. Cervical screening tests are now the only way to detect the disease in the absence of symptoms.

HPV infection is found in around 80% of low-grade squamous intraepithelial lesion (LSIL) cases and 90% of high-grade squamous intraepithelial lesion (HSIL) cases. In the UAE, the crude incidence rate of HPV-related cancer is 4.03 per 100,000 people. However, there is a lack of clear data on the prevalence of low and high-risk HPV in the country's general population. Multiple HPV genotypes have been observed occurring in the same individual, as documented [13,14]. This could involve a combination of HR genotypes together or HR and LR genotypes. Identifying the HPV genotype among infected individuals is essential for raising awareness about HR and LR genotypes. Implementing routine screening for at-risk individuals can help reduce the incidence of malignancy.

Data on the prevalence of HR and LR HPV genotypes associated with cervical cancer progression in the UAE remain limited. The emergence of new HPV genotypes not covered by the currently available 9-valent vaccine presents a potential public health concern, as these genotypes may continue to circulate and pose an increased risk if left undetected and untreated. Importantly, many of these emerging genotypes cannot be reliably identified using conventional diagnostic methods. Therefore, comprehensive molecular surveillance using real-time PCR is essential to ensure accurate detection of circulating HPV genotypes, especially the PNA-based fluorescence melting curve analysis in real-time PCR utilizes peptide nucleic acid probes that possess high target specificity, enabling precise hybridization to their complementary sequences, and high specificity allows the accurate detection of individual targets and supports multiplex analysis, making it possible to identify multiple HPV genotypes within a single reaction. Even a very minimal HPV load, as low as one molecule per $10^5$ cells, can be detected to support early intervention and prevention strategies [15].

This study explores the genotypic and molecular landscape of HPV infections among women in the United Arab Emirates, emphasizing the critical role of genotypic surveillance in preventing, detecting, and managing HPV-related diseases while promoting stronger public health initiatives.

## Methods

### Ethical approval

The present study received ethical approval from the Institutional Review Board (IRB) of Gulf Medical University, Ajman, United Arab Emirates (Approval Reference: IRB-COHS-FAC-79-DEC-2023; dated January 9, 2024). All study procedures, including design, data collection, participant selection, and protocol implementation, were conducted in strict accordance with the ethical principles outlined in the World Medical Association's Declaration of Helsinki (DoH–October 2013) and the 2023 Good Clinical Practice (GCP) guidelines of the National Drug Abuse Treatment Clinical Trials Network. Furthermore, the study upheld the ethical provisions of the Helsinki Declaration and ensured full compliance with patient information confidentiality throughout all stages of the research. Relevant research data and supporting literature were systematically obtained from PubMed, Scopus, reference lists of pertinent articles, and other reputable scientific databases.

### Study design

A retrospective cross-sectional study was conducted using liquid-based cytology (LBC) samples submitted from hospitals across multiple regions of the UAE, including the provinces of Ajman, Sharjah, Dubai, Umm Al Quwain, and Fujairah, for HPV screening and associated cytological evaluation. All samples were processed at the Department of Pathology and Microbiology, Thumbay Laboratory, Thumbay University Hospital (TUH), Ajman, UAE.

### Data collection

In accordance with the IRB approval (dated January 9, 2024), the data for the study samples were retrospectively collected from the date 10/01/2024 onwards from the medical records of Thumbay Laboratory, TUH, for the period of 12

months (10/01/2024 to 10/01/2025). The current study confirmed that the data of the study population from the medical records were fully anonymized before they were utilized for the study, excluding age, ethnicity, and nationality. The informed consent of the study population was waived by the IRB committee since it is a retrospective cross-sectional study. All collected LBC samples were reviewed for completeness of demographic, clinical, and laboratory information before inclusion. Patient age, nationality, cytology findings, and HPV results were cross-verified with hospital records. Samples with invalid or inconclusive HPV genotyping were reprocessed once and, if still unresolved, excluded from genotype-specific analysis. Missing non-critical variables were assessed for randomness, and pairwise deletion was used when appropriate; no statistical imputation was performed for HPV status or genotype due to their biological specificity. Potential confounders such as age, nationality, and cytological abnormalities were controlled using stratified analyses and multivariable logistic regression. Standardized laboratory protocols and uniform inclusion criteria were implemented to minimize variability across participating centers

## Study populations

A total of 229 LBC samples were included in this study. The sample size was determined by the availability of all eligible cervical samples received during the study period from women aged 20–55 years attending the Gynecology Outpatient Department at TUH, and other hospitals across the Northern Emirates of the UAE for HPV screening. Because this was a retrospective, laboratory-based study, all samples submitted for primary HPV screening and cytological evaluation were included to ensure complete representation of the screened population. This approach allowed for maximal capture of real-world screening data and enhanced the reliability of genotype distribution analysis. Patients with a prior history of HPV infection, cervical cancer, or those undergoing follow-up testing were excluded to avoid selection bias and to ensure that the study cohort reflected individuals undergoing initial HPV screening. Thus, the sample size of 229 represents a comprehensive census of all eligible cases during the defined study period rather than a calculated sample size. The sample collection was performed by qualified gynecologists using the SurePath® Liquid-Based Cytology System (TriPath Imaging, Burlington, NC, USA). Upon receipt, all samples were processed and examined cytologically in accordance with the laboratory's standardized protocols [16].

## Cytodiagnostic investigation

Papanicolaou-stained Thin Prep liquid-based PAP smears (Thin Prep 2000 tissue processor, Hologic, Inc., MA, USA) were reviewed by qualified pathologists to confirm the diagnosis of cytological abnormalities. The diagnosed slides were classified based on morphologic criteria for cervical neoplasia according to the Bethesda System 2014 [16]. All PAP smear slide results and demographic data (nationality and age) of the patients were retrieved from the medical records of Thumbay Laboratory, TUH.

## HPV genotyping

The residual concentrated materials in the Thin Prep concentrated tubes were used for HPV DNA extraction, followed by the detection of high and low-risk HPV genotypes using the PNA-based RT-PCR method [17].

## Extraction of HPV DNA by magnetic beads

Residual ThinPrep samples were first centrifuged at 3000 rpm for 10–15 minutes, and the resulting pellet was subjected to lysis using lysis buffer and Proteinase K, followed by incubation at 55°C. RNase treatment was then performed at room temperature to remove residual RNA. DNA extraction was carried out using a magnetic bead–based method, in which the lysate was mixed with magnetic beads and sequentially washed with DNA wash solutions 1 and 2. The extraction process was automated using the Genolution Nextractor® NX-48S (Genolution Nextractor® NX-48S, Gangseo-gu, Seoul, South

Korea) system to ensure efficient magnetic separation and purification [18]. Finally, DNA was eluted at 65°C, separated from the magnetic beads, and stored at –20°C for subsequent real-time PCR analysis [19].

## Sample preparation for real-time PCR

For HPV genotyping, 5 µl of extracted DNA was aliquoted into three reaction tubes (A1, B1, and O1), each supplemented with 19 µl of the corresponding HPV reagent mix (Mix A, Mix B, or Mix O) and 1 µl of Taq DNA polymerase, yielding a final reaction volume of 25 µl. Positive and negative controls for each mix were included in every run. Reactions were sealed, briefly centrifuged, and processed on the CFX96™ Real-Time PCR System (Bio-Rad, USA) following the manufacturer's protocol for the PANA RealTyper™ HPV 32 Genotyping Kit (PANAGENE, South Korea). Fluorescence signals (FAM, HEX/ VIC, ROX, and Cy5) were monitored and melting curve thresholds were set according to the manufacturer's instructions to accurately determine genotype-specific HPV detection [17]. The baseline cut-off of the melting peak analysis of each fluorescent dye and HPV DNA detection criteria is provided in S1 and S2 Tables and also represented in Fig 1.

## Statistical analysis

Data were analyzed using IBM SPSS for Windows Version 28.0 (IBM Corp., Armonk, NY, USA). Descriptive statistical tests were conducted, and the categorical data are presented with frequency and percentage. A Chi-square test was conducted in Table 2 to determine the association between the dependent variable, HPV status, and the independent variables: age group, nationality, and genotypes, with $p$-value of <0.05 considered statistically significant. Binary logistic regression was conducted in Table 4 to determine the predictors of positive HPV status, including age groups and nationalities.

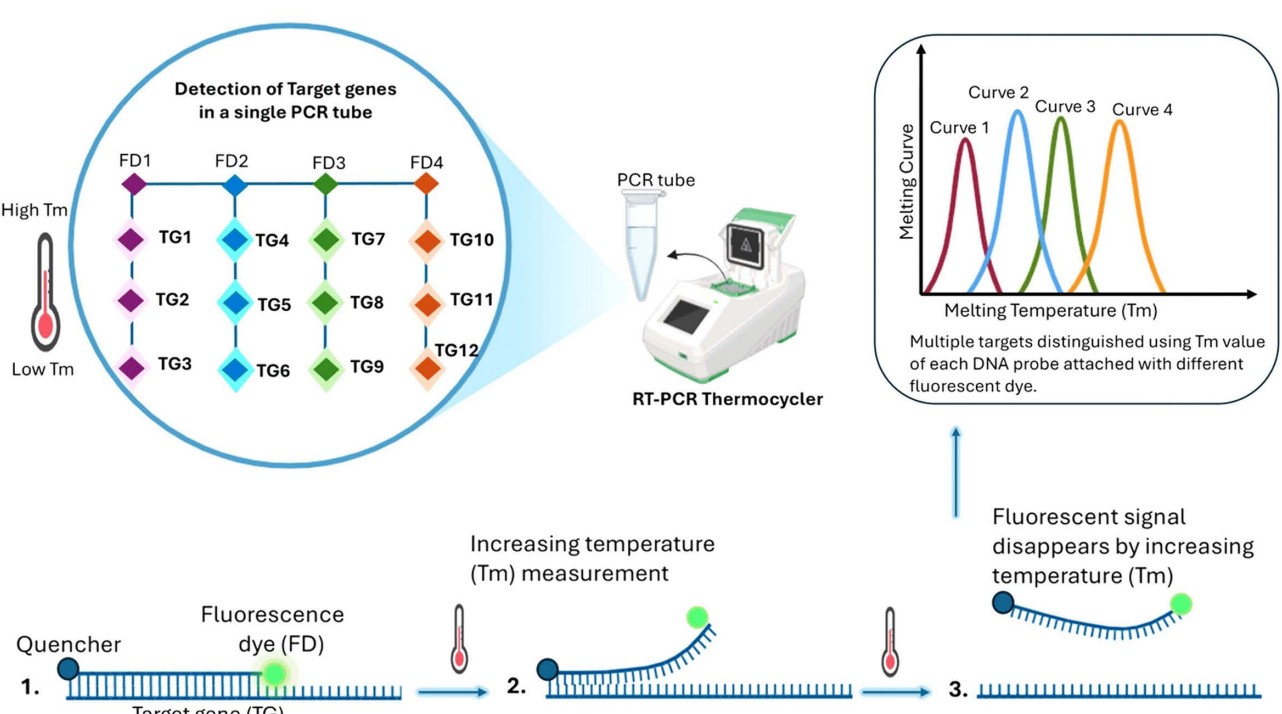

**Fig 1. Diagrammatic representation of the detection of HPV genotypes using peptide nucleic acid-based fluorescence melting curve analysis.**

## Result

### Cytodiagnostic Investigation

A total of 229 Pap smear samples were processed for routine cytopathological evaluation to assess cervical epithelial changes. According to the Bethesda 2014 system, 20 samples (9%) were classified as LSIL, 39 (17%) as ASCUS, 2 (1%) as AGC, and 1 (0.4%) as ASC-H. The majority of samples, 167 (73%), were reported as NILM. Their details are presented in S3 Table and Fig 2–3.

### Identifying high and low-risk HPV

Of the 229 Pap smear samples analyzed, 96 (42%) were HPV-positive, and 133 (58%) were HPV-negative. A total of 191 HPV genotypes were identified in the 96 positive samples, including 116 genotypes from 47 abnormal cytology cases (ASCUS = 26, LSIL = 18, AGC = 2, ASC-H = 1) and 75 genotypes from 49 NILM samples. In ASCUS samples, 61 HPV genotypes were detected, comprising 39 HR and 22 LR types; HR53 & 16 were the most frequent. The 18 LSIL samples

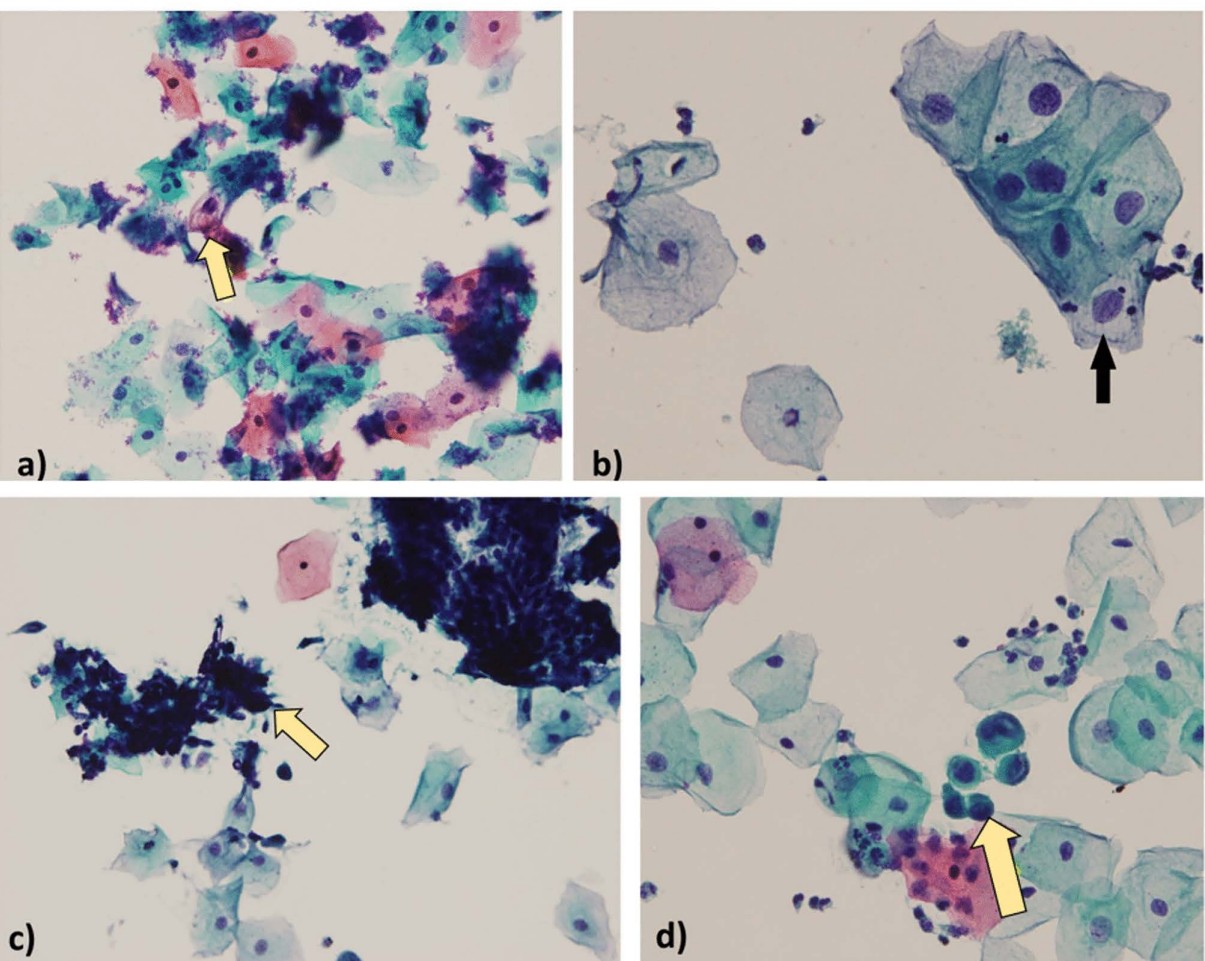

**Fig 2. Grading of PAP smeared cytological samples (400X magnification, Olympus BX51).** a) Atypical squamous cells of undetermined significance (ASCUS), b) Low-grade squamous intraepithelial lesion (LSIL), c) Atypical glandular cell (AGC), d) Atypical squamous cells, cannot rule out high-grade squamous intraepithelial cells (ASC-H).

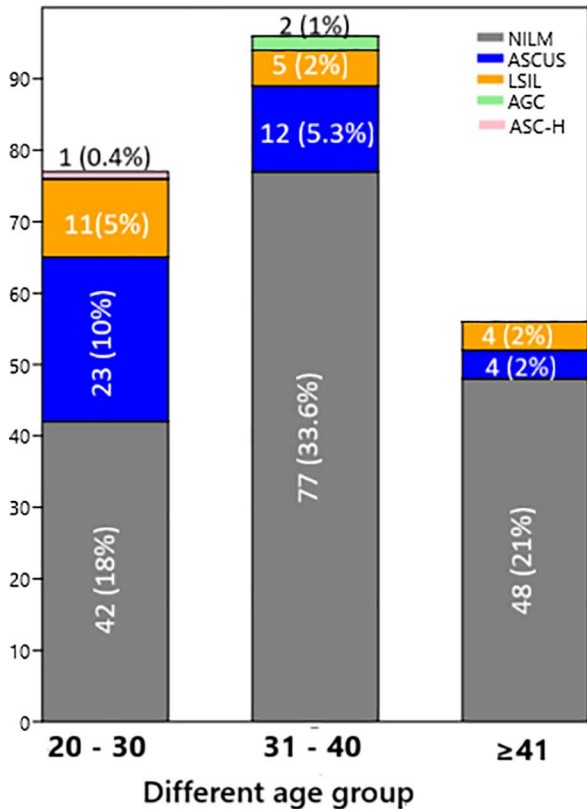

**Fig 3. Cytology grading of PAP smear samples collected from different age groups of Women.**

yielded 51 genotypes (34 HR, 17 LR), with HR53 & 66 predominating. In AGC cases, HR51, 31, and LR81 were identified, while HR68 was detected in the ASC-H sample. Among NILM samples, HR16 was the most common, followed by HR18, 68, 53 & 31, with LR6, 81 & 61 also detected. The distribution of single and multiple H & LR-HPV infections across cytological categories is summarized in S3 and S4 Tables, and in Fig 4–5.

### Age and ethnic diversity in HPV-positive study cohort

The study population comprised three age groups: 20–30 years (42; 44%), 31–40 years (40; 42%), and ≥41 years (14; 14%). The details are given in the S3 Table. Overall, 114 (50%) participants were Arab, and 115 (50%) were non-Arab. HPV positivity was observed in 39 Arabs (17%) and 57 non-Arabs (25%), while 75 Arabs (33%) and 58 non-Arabs (25%) tested HPV-negative. Detailed demographic characteristics, HPV positivity, and distribution of single or multiple-infection patterns with H and LR genotypes are presented in S4 Table. A total of 80 (42%) HPV genotypes were detected in women aged 20–30 years, with HR16 being the most frequent, followed by HR68 & 53. In the 31–40-year age group, 87 (46%) genotypes were identified, predominantly LR6 and HR31, 53, 35 & 66. Among women aged ≥41 years, 24 (12%) genotypes were detected, mainly HR16, 53, 31, 53, 66, and 59.

Ethnicity-based analysis showed that in the 20–30-year group, 37 HPV genotypes were detected among Arabs, with HR53 and HR16 predominating, whereas 43 genotypes were identified among non-Arabs, with HR68 and HR51 being most common. Similar distributions were observed in the 31–40-year group, with HR53 and LR6 prevailing among Arabs and HR66, 51, and 31 among non-Arabs. Fewer genotypes were detected among women aged ≥41 years, with a

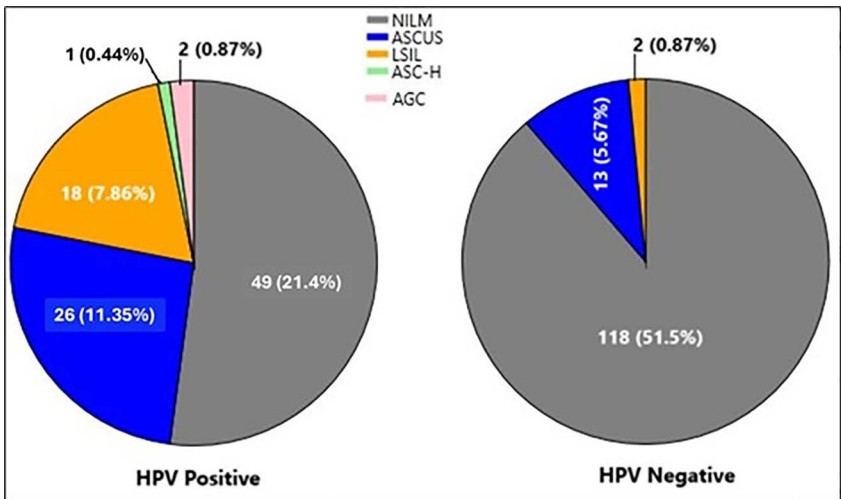

**Fig 4. Details of cytological grading of PAP smear samples with HPV positive and negative status.**

sporadically distributed pattern across both ethnic groups. The details are presented in Fig 6. Of the 96 HPV-positive samples, 46 (48%) showed single-genotype infections (30 HR and 16 LR), while 50 (52%) had multiple-genotype infections, including multiple HR (24%), multiple LR (2%), and mixed HR/LR infections (26%) and the details are mentioned in the S4 table. The most prevalent genotypes were HR53 (7%), HR16 & 68 (6% each), HR66 & 31, and LR6 (5% each), followed by HR35, 45 & LR61 (4% each), HR18 & 51, and LR43 (3% each), and the details are provided in Table 1.

A chi-square analysis was performed to examine the association between HPV status and selected demographic and clinical variables, including age group, nationality, and HPV genotype. A statistically significant relationship was observed between age and HPV positivity ($\chi^2 = 11.63$, $p = 0.003$). Participants aged 20–30 years demonstrated the highest prevalence of HPV infection (54%), with a progressive decline in positivity across increasing age groups, reaching the lowest rate among individuals ≥41 years (25%). Nationality also showed a significant association with HPV status ($\chi^2 = 4.93$, $p = 0.026$), wherein non-Arab participants exhibited a higher HPV positivity rate (50%) compared to Arab participants (34%). The details are mentioned in Table 2, and the nationality-wise HPV status among the study population is also provided in Fig 7. The most prevalent HPV genotypes were detected with their frequencies among the HPV positive women in the Arab and non-Arab study groups, and these details are presented in Table 3.

Women aged 31–40 have lower odds of having positive HPV compared to those aged 20–30, but the association is not statistically significant, Adjusted OR = 0.555 (95% CI: 0.299–1.031), $p = 0.063$. Individuals aged ≥41 are much less likely to experience the outcome than those aged 20–30, Adjusted OR = 0.304 (95% CI: 0.144–0.643), $p = 0.002$. Arabs have about 48% lower odds of the outcome than non-Arabs, with a statistically significant association (adjusted OR = 0.513, 95% CI: 0.295–0.990, $p = 0.017$). The details are given in Table 4.

## Discussion

In this study, the prevalence of HPV infection is lower (42%) than in previous studies conducted in the UAE in 2023 (61%), 2021 (56%), and 2018 (99%) [13,20,21]. However, this prevalence rate is much higher than in neighboring countries such as Oman (18%), Saudi Arabia (5% & 30%), and Qatar (31%) [22–25], and even higher than a study from China (16.5%) [26]. In our study, the prevalence of HPV infection among women of reproductive age with suspected pap smear abnormalities was 42%, which is lower than the rates reported in earlier studies from the UAE in 2023 (61%) & 2021 (56%) [13,20], However, the current prevalence of HPV is lower than the previous reports conducted across various countries in

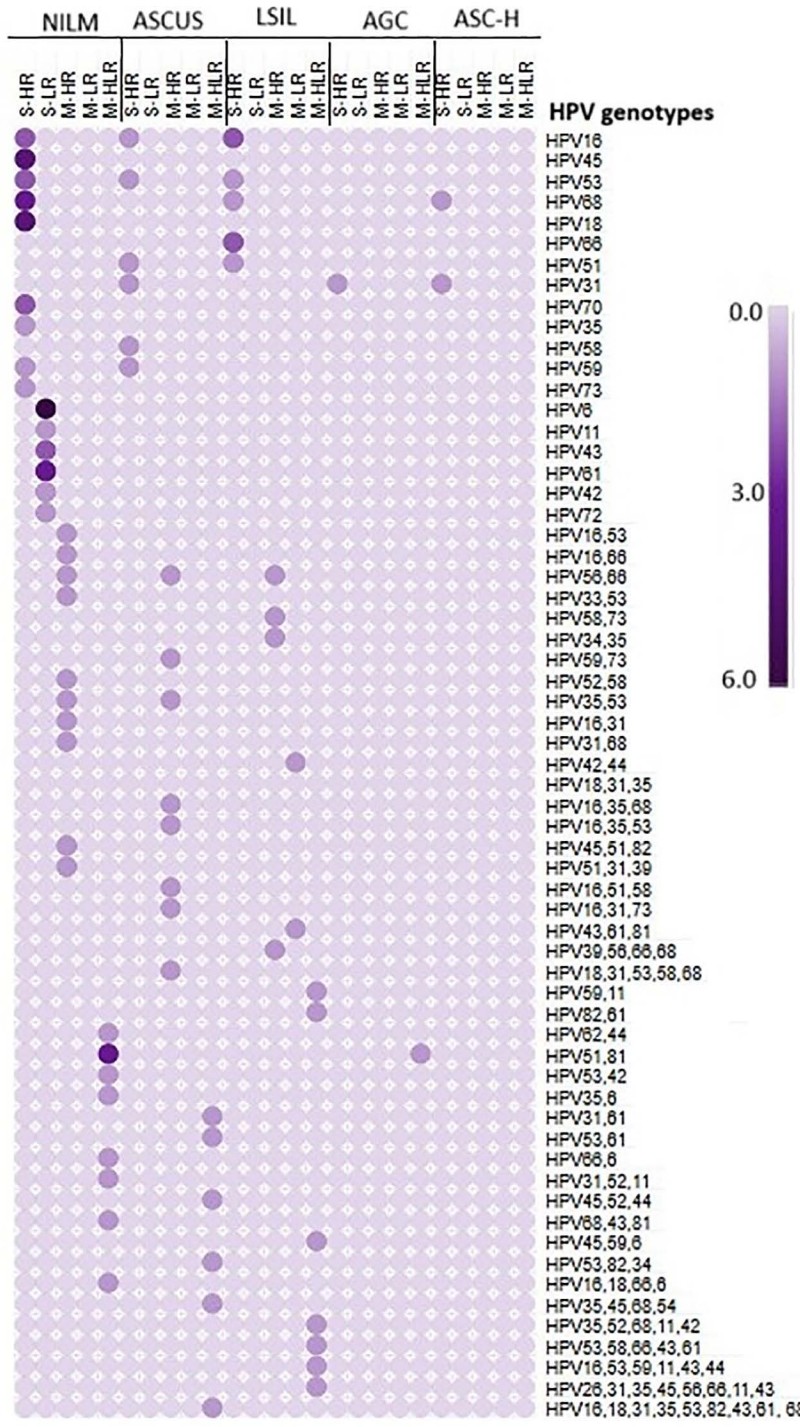

**Fig 5. Number of single- and multiple-, low- and high-risk genotypes detected in different cytology samples.**

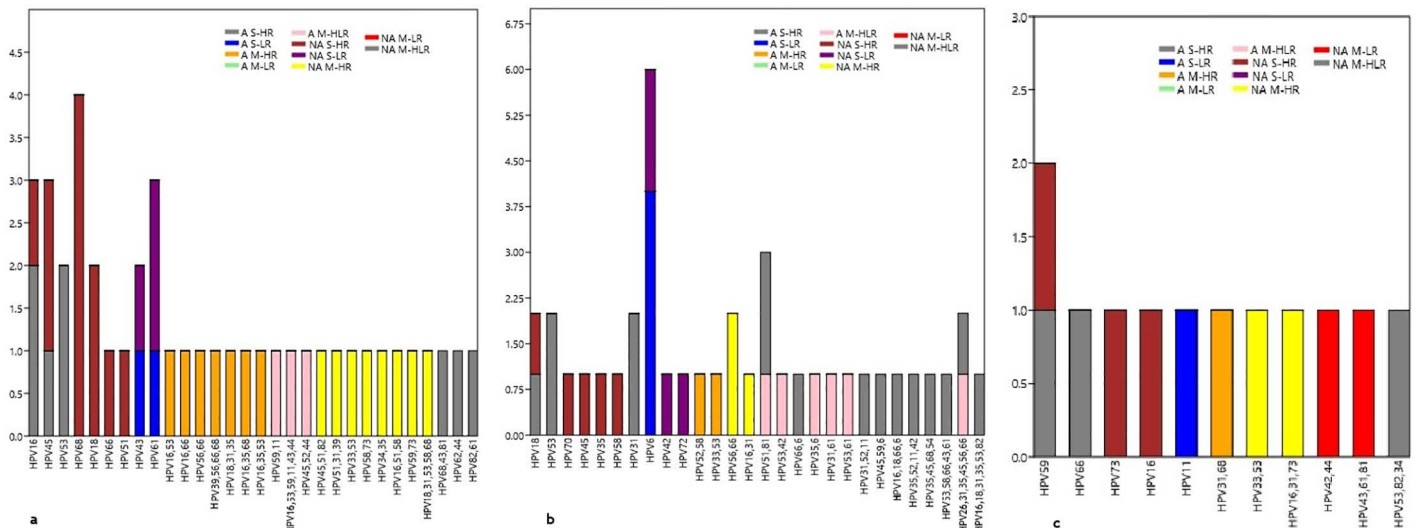

**Fig 6. Frequency of single, multiple low and high-risk HPV genotypes among the different age groups of Arab and non-Arab study population.**
a) Age group 20–30; b) 31–40; c) ≥41.

**Table 1. Number of high and low-risk genotypes detected among the HPV positive samples.**

| S. No | No. study population *n* = 229 | | | |
|---|---|---|---|---|
| | Total number of genotypes detected in HPV Positive cases (*n* = 96; 42%) *n* = 191;83% | | | |
| | High-risk HPV Genotype *n* = 137 (60%) | | Low-risk HPV Genotype *n* = 54 (23%) | |
| | HPV Genotype | No. (%) | HPV Genotype | No. (%) |
| 1. | 53 | 16 (7) | 6 | 11 (5) |
| 2. | 16 | 13 (6) | 61 | 9 (4) |
| 3. | 68 | 13 (6) | 43 | 8 (3) |
| 4. | 66 | 12 (5) | 11 | 6 (3) |
| 5. | 31 | 11 (5) | 81 | 6 (3) |
| 6. | 35 | 10 (4) | 42 | 4 (2) |
| 7. | 45 | 9 (4) | 44 | 4 (2) |
| 8. | 18 | 8 (3) | 34 | 2 (1) |
| 9. | 51 | 8 (3) | 70 | 2 (1) |
| 10. | 59 | 6 (3) | 72 | 1 (0.5) |
| 11. | 58 | 6 (3) | 54 | 1 (0.5) |
| 12. | 52 | 6 (3) | – | – |
| 13. | 82 | 5 (2) | – | – |
| 14. | 73 | 4 (2) | – | – |
| 15. | 56 | 4 (2) | – | – |
| 16. | 33 | 3 (1) | – | – |
| 17. | 39 | 2 (1) | – | – |
| 18. | 26 | 1 (0.5) | – | – |

**Table 2. Association of age groups, nationality, and genotypes with HPV status.**

| Variables | Categories | HPV status n = 229 | | Chi-square | p-value |
|---|---|---|---|---|---|
| | | Positive (96) n (%) | Negative (133) n (%) | | |
| Age Group | 20–30 | 42 (54) | 35 (46) | 11.63 | 0.003* |
| | 31–40 | 40 (42) | 56 (58) | | |
| | ≥41 | 14 (25) | 42 (75) | | |
| Nationality | Arabs | 39 (34) | 75 (66) | 4.93 | 0.026* |
| | Non-Arabs | 57 (50) | 58 (50) | | |

n – No. of participants, *- statistically significant differences.

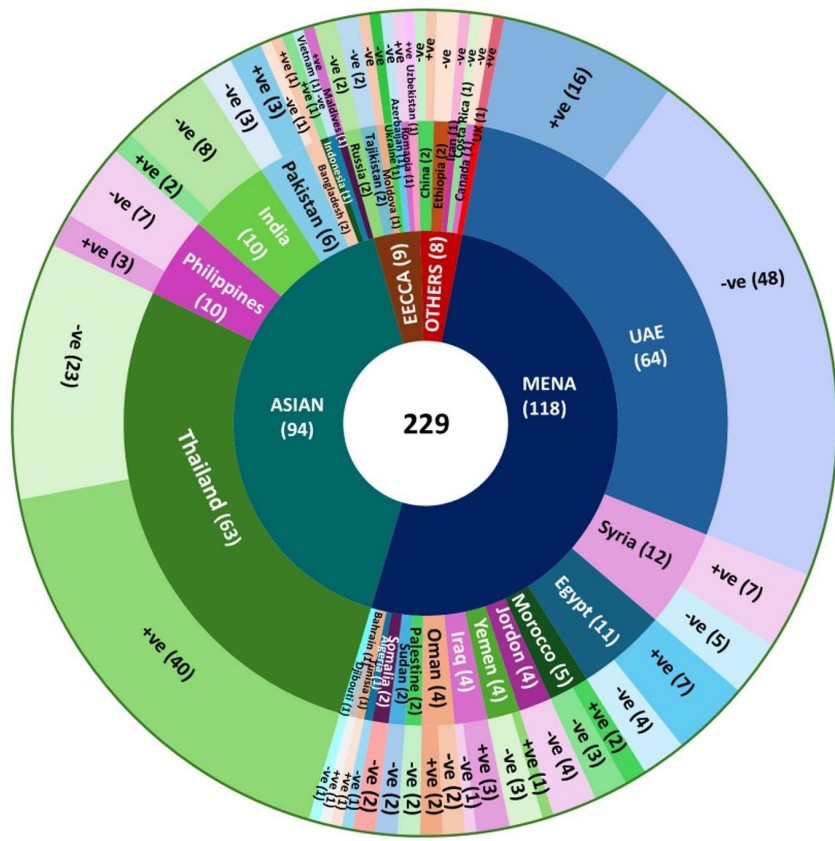

**Fig 7. Nationality-wise HPV status among the study population.**

the MENA region (Egypt (76%), Turkey (75%), Lebanon (65%), Syria (61%), and KSA (53%)) [27–31]. In contrast, a markedly lower prevalence of 18% has been documented in Oman [22].

Notably, the prevalence of abnormal cervical cytology in the UAE has risen by nearly 60% when compared with earlier reports by Ortashi and Abdalla, Fakhreldin and Elmasry, and Al-Zaabi et al. [32–34]. Whereas, a report from China stated that 93.5% of Pap smear samples exhibited abnormal cytology, Bitarafan et al. from Iran found that 73% of Pap smear

**Table 3. Total number of the most prevalent HPV genotypes among the HPV-positive women, and their ethnicity.**

| HR/LR genotypes | No. of HPV genotypes n | Total No. of study population n = 229 | |
|---|---|---|---|
| | | No. of HPV positive cases (n = 96; 42%) | |
| | | Arab cohort (n = 40; 17.5%) n (%) | Non-Arab cohort (n = 56;24.5%) n (%) |
| HR-HPV | 53 (n = 16) | 11 (5) | 5 (2) |
| | 16 (n = 14) | 8 (3) | 6 (3) |
| | 68 (n = 13) | 3 (1) | 10 (4) |
| | 66 (n = 12) | 5 (2) | 7 (3) |
| | 31 (n = 12) | 6 (3) | 6 (3) |
| | 35 (n = 10) | 4 (2) | 6 (3) |
| | 18 (n = 8) | 3 (1) | 5 (2) |
| LR-HPV | 61 (n = 12) | 7 (3) | 5 (2) |
| | 6 (n = 9) | 4 (2) | 5 (2) |
| | 43 (n = 8) | 3 (1) | 5 (2) |
| | 11 (n = 6) | 3 (1) | 3 (1) |
| | 81 (n = 6) | 1 (0.4) | 5 (2) |

n – No. of genotypes, (%) – Prevalence

**Table 4. Predictors of Positive HPV status.**

| Variables | Categories | p-value | Crude OR (95% CI) | p-value | Adjusted OR (95% CI) |
|---|---|---|---|---|---|
| Age Group | 20-30 | – | – | – | – |
| | 31-40 | 0.091 | 0.592 (0.323–1.087) | 0.063 | 0.555 (0.299–1.031) |
| | ≥41 | 0.002 | 0.306 (0.146–641) | 0.002 | 0.304 (0.144–0.643)* |
| Nationality | Arabs | 0.017 | 0.522 (0.306–0.891) | 0.017 | 0.513 (0.295–0.990)* |
| | Non-Arabs | – | – | – | – |

*Binary Logistic Regression, references age group 20–30, nationality non-Arab.

samples demonstrated abnormal cytological findings [26,35]. In contrast, a study from Thailand reported that 34% of the PAP smear samples showed abnormal cytology [36].

In this study, 76% of abnormal cytological samples were positive for HPV infection, whereas 29% of NILM samples showed positive for HPV infection. In 2021, a study from KSA mentioned that 57% were positive for HPV infection in the abnormal cytological samples, whereas 43% were positive among normal epithelia [23].

The overall prevalence of HR genotypes was higher (60%) than that of LR genotypes (23%) in this study. A similar pattern of HR prevalence was observed in many previous studies conducted in Oman, KSA, Thailand, and Iran [22–24,35,36].

In this study, the highest prevalent genotype was HR53, followed by HR16, 68, 66, 31 & LR6, then HR35, 45 & LR61, and HR18, 51 & LR43, then LR11, 81, HR58, 59 & 52 respectively. Various studies conducted in the UAE showed that there is no uniform occurrence of HPV genotypes, but HR16 and 31 were the predominant genotypes [13,20,21,37]. But in 2023, the LR6 was the most dominant genotype, after that HR16 [13]. In contrast, in the current study, HR53 was dominant, and then HR16 & 68. Similarly, Sanjos et al in 2007 stated that the HR53 was the highest prevalent genotype in

Eastern Africa and Central America, and in 2010, Bruni et al noted that HR53 was the most predominant genotype in the African and North American continents, but worldwide, the prevalence rate of HR53 was 0.6% [38,39]. Similarly, Osmani et al found that HR53 & 16 were the dominant genotypes globally, but in Asia, 52 & 53 were dominant, and a study from China observed that HR53 was one of the second most predominant genotypes along with 52 [40,41].

A higher HPV positivity rate (36%) was observed among the participants aged 20–40 years in the current study. This finding is consistent with a 2019 study conducted in Saudi Arabia, which also reported the highest HPV prevalence in the same age groups [42]. A study from Bahrain also confirmed that a higher rate of HPV positivity was detected in the age group of 31–40 years females [43], and the same result was found in the reproductive age group (21–40 years) of the study population in Iraq [44]. However, in contrast to our study, Al-Shammari et al observed higher HPV positivity in the age group of 40–50 years females, and the same result was found in the same age group of the female study population of China [23,26].

The present study found variations in HPV genotype distribution among the Arab and non-Arab study cohorts. Among the Arab study population, the most encountered genotype was HR53, followed by HR16, LR61, and HR31 & 66, respectively. Among them, the HR 53 was dominant in the UAE nationalities, followed by Iraqi participants. Whereas, among the non-Arab cohort comprising Asian and Central Asian, HR68 was the most frequently observed, followed by 66, 16, 31& 35, and then HR18 & LR61, 6, 43 & 81. In this cohort, the HR68 & 66 were predominantly identified in the Thai study population. Overall, the genotype HPV53 is emerging among the Arab study group, whereas the HR68 was increasing among the non-Arab cohort, of which the Thailand study participants show predominance. The occurrence of HR18 in this study was lower among the Arabs than the non-Arab study group, which could be due to the implementation of the HPV vaccine program in the MENA region. There was an equal distribution of HR31 & LR11 occurrences in both study cohorts. In general, among both study groups, the HPV genotype frequencies are more commonly seen among the Thai participants. Similarly, a low positivity rate of HPV18 was detected in the Western Iranian and Egyptian study group [35,45].

In the mixed co-infection of HPV genotypes, HR 53, 16, 66, 31, 35, and 68 were more frequently encountered than the LR genotypes (61, 81, 6, 11). 23% of PAP smear samples were infected by ≥3 different genotypes, of which the non-Arab cohort showed a higher prevalence of mixed co-infection (16%) than the Arab cohort (7%). Among the mixed co-infection cases, 32% of the study population were found to be NILM, of which 3 cases exhibited 6–9 different genotypes. Likewise, 8% of the study population had mixed genotypic infection in the Iraqi study population [44]. Similarly, Obaid et al noted that 7% of the study population were infected with mixed HPV genotypes. However, the study disclosed differences between Saudi and non-Saudi nationalities. [46].

In the previous study conducted by Odeh et al., several HPV genotypes were observed in combined form, such as 62/81, 6/43, 31/68, 44/55, and 52/86 [13]. These combination results were due to limitations of the reverse hybridization diagnostic method, which did not allow for precise differentiation of individual genotypes. Whereas the current study accurately identified each genotype independently using an enhanced diagnostic approach, confirming individual genotypes such as LR81, 6, 43, and 44, as well as HR31, 68, and 52.

Our analysis demonstrated that specific HPV genotypes, particularly HR53 (11; 5%) &16 (8; 3%) and LR61 (7; 3%), were more commonly detected among the Arab study group originating from MENA-region countries residing in the UAE. In contrast, genotypes HR68 (10; 4%), 66 (7; 3%) & 35 (6; 3%) LR81 (5; 2%) were more frequently observed among the Asian study group. This Asian cohort included a substantial number of participants from Thailand, where genotypes HR68 (3%), HR66 (3%), HR35 (2%), and LR81 (2%) were detected.

The UAE was the first in the Eastern Mediterranean Region to incorporate the HPV vaccine into the national immunization program for females in 2018. In 2023, the program was expanded to include males aged 13–14 years, thereby strengthening nationwide prevention strategies against HPV-related diseases in both genders. By 2030, the UAE aims to achieve 90% HPV vaccination coverage among girls before the age of 15 [47].

The findings of the present study highlight that several of the predominant genotypes identified (HR53, 68, 66, 35, and LR61, 43, 81) are not included in the currently available 9-valent HPV vaccine (genotypes 6, 11, 16, 18, 31, 33, 45, 52,

and 58). Observed differences in HPV genotype distribution between Arab and non-Arab women in this study may reflect underlying socio-behavioral and healthcare-access disparities in the UAE. Variations in health-seeking behavior, socioeconomic mobility, and access to gynecological services likely further influence these differences.

These findings highlight the awareness strategies and improved access to screening and HPV vaccination across all population groups. Despite the implementation of a national vaccination program, monitor unvaccinated populations and promote the importance of HPV immunization in reducing HPV-related cancer burden. Furthermore, continued surveillance through HPV screening using real-time PCR for women aged 25 years and above is essential to ensure timely detection of circulating HPV genotypes. Overall, the study underscores that several prevalent genotypes in this population are not covered by existing HPV vaccines, emphasizing the importance of considering region-specific genotype distribution in future vaccine development and public health strategies.

## Conclusion

The present study demonstrates that HR53, 16, 68, 66, 31, 35, 45, and 18, along with LR6, 61, 43, 11, and 81, were the most prevalent HPV genotypes detected among HPV-positive Pap smear samples from women aged 20–55 years. Mixed infections were identified in 26% of HPV-positive cases, involving combinations of high- and low-risk genotypes, with HR53 emerging as the predominant genotype, followed by HPV35, 66, 11, 43, 81, 61, and 6. Notably, 21% of NILM samples tested positive for HPV and harbored diverse high- and low-risk genotypes, indicating the presence of subclinical infection within cytologically normal samples. A higher HPV positivity rate was observed in the non-Arab cohort compared with the Arab cohort. Genotype-specific differences were also evident: HR53 appeared to be emerging predominantly in the Arab population, whereas HR68 showed increased prevalence within the non-Arab group, particularly among Thai participants. HR18 was less frequent among Arab women.

Collectively, these findings provide clinically meaningful insights into the evolving HPV genotype landscape in the UAE and offer essential baseline data for future longitudinal research, vaccine effectiveness evaluations, and genotype-targeted screening strategies. The study underscores the need for national stakeholders to reinforce existing HPV screening policies and to account for region-specific genotypes not covered by the current 9-valent vaccine. The prominence of non-vaccine HPV genotypes in this population highlights the importance of considering their inclusion in future vaccine formulations. Integrating such evidence into public health planning will strengthen cervical cancer prevention efforts and support alignment with global initiatives aimed at cervical cancer elimination.

## Limitations of the study

This study provides one of the most detailed recent assessments of HPV genotype distribution among women undergoing cervical cytology testing in the UAE. A key strength lies in the use of real-time PCR, which enhances genotypic resolution and reduces cross-reactivity between H&LR-HPV types. The inclusion of samples from multiple healthcare facilities and strict adherence to uniform laboratory protocols further strengthened internal validity and reduced variability in sample processing. However, several limitations should be acknowledged. First, the study relied on samples submitted for routine clinical testing, which may introduce sampling bias, as women attending gynaecological clinics may not represent the broader population of the UAE, particularly those who are asymptomatic or have limited health insurance coverage. Second, incomplete demographic details such as marital status, sexual history, and vaccination status, or clinical data in some cases, necessitated the exclusion of certain variables, which may have reduced statistical power.

## Supporting information

**S1 File. For each variable of interest, give sources of data and details of methods of assessment (measurement).** Describe the comparability of assessment methods if there is more than one group. (DOCX)

**S2 File. (a) Report numbers of individuals at each stage of study – e.g., numbers potentially eligible, examined for eligibility, confirmed eligible, included in the study, completing follow-up, and analysed.**
(DOCX)

**S3 file. Flow diagram of the study.**
(DOCX)

**S4 File. Cross-sectional study—Report numbers of outcome events or summary measures.**
(DOCX)

**S5 File. Cautious overall interpretation of results considering objectives, limitations, multiplicity of analyses, results from similar studies, and other relevant evidence.**
(DOCX)

**S1 Checklist. STROBE-checklist-v4.**
(DOCX)

**S1 Table. Baseline cut-off (threshold) of melting peak analysis of each fluorescent dye in each HPV tube.**
(DOCX)

**S2 Table. Criteria for target HPV detection.**
(DOCX)

**S3 Table. Detection of HPV infection in different cytology grading in various age groups of the study population.**
(DOCX)

**S4 Table. Number of single, multiple low and high-risk HPV genotypes in different cytology samples and different age groups of the study population.**
(DOCX)

## Acknowledgments

We would like to acknowledge the efforts of Pathologists, Microbiologists, Gynaecologists, and Molecular biologists, and other relevant staff of Thumbay Laboratory and Thumbay University Hospital, and other hospitals and clinics across the UAE for sending samples for diagnosis of PAP smear and HPV status towards this research. Also, thank the IRB, Gulf Medical University, for approval of the research. We express our gratitude to the respected higher authorities of the College of Health Sciences and the College of Medicine for their constant support in conducting this research.

## Author contributions

**Conceptualization:** Nazeerullah Rahamathullah.

**Data curation:** Devapriya Finney.

**Formal analysis:** Nazeerullah Rahamathullah, Devapriya Finney, Sunil Kumar Bylappa, Ameena Ebrahim Mugar, Mohammed Abdulrazzaq Jabbar.

**Investigation:** Devapriya Finney, Sunil Kumar Bylappa.

**Methodology:** Nazeerullah Rahamathullah, Devapriya Finney, Sunil Kumar Bylappa, Ameena Ebrahim Mugar.

**Project administration:** Nazeerullah Rahamathullah, Manivannan Nandhagopal.

**Resources:** Sunil Kumar Bylappa, Ahmed Luay Osman Hashim, Manivannan Nandhagopal.

**Software:** Nazeerullah Rahamathullah.

**Supervision:** Nazeerullah Rahamathullah, Ahmed Luay Osman Hashim, Manivannan Nandhagopal.

**Validation:** Nazeerullah Rahamathullah.

**Visualization:** Nazeerullah Rahamathullah, Ameena Ebrahim Mugar, Ahmed Luay Osman Hashim.

**Writing – original draft:** Nazeerullah Rahamathullah, Devapriya Finney.

**Writing – review & editing:** Nazeerullah Rahamathullah, Devapriya Finney, Ahmed Luay Osman Hashim, Manivannan Nandhagopal.

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
