## [Decision Letter · Decision Letter 0]

8 Oct 2025

Dear Dr. Rahamathullah,

We look forward to receiving your revised manuscript.

Kind regards,

Ivan Sabol

Academic Editor

PLOS ONE

2. Please include a separate caption for each figure in your manuscript.

Additional Editor Comments:

There is significant textual overlap between the current study and the preceding study (reference 13).

Odeh et al. (2023) Exploring the prevalence of Human Papillomavirus (HPV) genotypes in PAP smear samples of women in northern region of United Arab Emirates (UAE): HPV Direct Flow CHIP system-based pilot study. PLOS ONE 18(9): e0286889. https://doi.org/10.1371/journal.pone.0286889

Some paragraphs are identical in both manuscripts. The presentation of several figures and tables appears to be substantially identical as well. Beyond removing the technical overlap, it is also necessary to provide some justification as to why the new (replication) study was needed as per PLOS criteria for publication subheading “Replication Studies” https://journals.plos.org/plosone/s/criteria-for-publication (same goal, same population, same methodology except the last HPV typing assay step, same layout and presentation, only slight details in the conclusion with much of the text copy pasted verbatim).

Furthermore some issues to fix are:

P6 L 112-128. Study design/Data collection/Population sections. Are the 229 samples all samples collected in 12 months from multiple hospitals? Information about inclusion and exclusion criteria are missing. What were the reasons patients visited hospitals (line 113) was this screening or a referral population?

P8 L168-169 the assay information (RealTyperTM HPV 32 Genotyping kit) is not consistent with reference 17 provided for the method (“detecting 40 types”). Assay version should be disclosed in the methods section in case the company makes further changes to the product sold.

P8 L147 assay name doesn’t need to be repeated in the same paragraph

P9 results section. The study presents numerous highly overlapping tables (3) and figures (9) that often show high granularity of the data (ie Table 1 has 80 cells of data of which only 16 have 5 or more cases). Instead, consider including a supplementary excel table with detailed information and focus the results section only on critical findings. Also, it is very difficult to find the overall prevalence of individual types which are later referred to in the text.

Percentages should be rounded to 1 decimal place in the results text and tables.

P10 Table 1 has marginally different age grouping than previous study. Why was the grouping changed? From a statistical point, 51>category is too small compared to other categories and could have been combined with 41-50 one. Alternatively, age groping of previous large scale reference studies should be used and the current data compared to those (ie later mentioned Bruni et al 2010)

P14 L237 the figure 6 should be replaced by a table (or at least the summary information provided as a supplement table) to make it more readable

P17 L302-306 the p value was calculated on the total number of positive and negative cases. This is not shown on Table 3 where the same p value details are shown.

P18 Discussion. Critically, unless the type of population studied (general screening, referral, convenience) is known, any comparisons with previous studies is impossible to appreciate or understand.

The prevalence data are not put in the context of the world in general. For example Bruni L et al, Cervical Human Papillomavirus Prevalence in 5 Continents: Meta-Analysis of 1 Million Women with Normal Cytological Findings, The Journal of Infectious Diseases, Volume 202, Issue 12, 2010, Pages 1789–1799, https://doi.org/10.1086/657321 or similar encompassing study.

The most commonly found HPV53 was not put into sufficient context in the discussion or conclusion

The discussion doesn’t sufficiently address the differences between Arab and non-Arab participants and thus doesn’t clearly state whether any differences are notable or is such differentiation irrelevant for future studies

Supplementary file S2 doesn’t answer the question of potentially eligible participants. The file completely overlaps the manuscript text and can be removed as uninformative or revised

Reviewers' comments:

Reviewer's Responses to Questions

**Comments to the Author**

1. Is the manuscript technically sound, and do the data support the conclusions?

Reviewer #1: Yes

Reviewer #2: Yes

2. Has the statistical analysis been performed appropriately and rigorously?

Reviewer #1: Yes

Reviewer #2: No

3. Have the authors made all data underlying the findings in their manuscript fully available?

Reviewer #1: Yes

Reviewer #2: No

4. Is the manuscript presented in an intelligible fashion and written in standard English?

Reviewer #1: Yes

Reviewer #2: Yes

Reviewer #1: The authors tried to highlight the geographic distribution of HPV genotypes among PAP smear samples by using melted PCR technology. Their finding underscores the importance of molecular testing for HPV genotyping.

one Commnet to highlight is the distinction of the samples as Arab and non-Arab is a crude data i.e. it won't give a casual inference as the details of these population is not known including the epidemiological profile, sexual history etc.

Reviewer #2: Summary:

This manuscript presents a retrospective cross-sectional study investigating the genotypic distribution and molecular epidemiology of human papillomavirus (HPV) among women in the United Arab Emirates (UAE). Using a peptide nucleic acid (PNA)-based fluorescence melting curve analysis via real-time PCR, the authors analyzed 229 PAP smear samples collected from several hospitals between 2024 and 2025. The study identified high- and low-risk HPV genotypes, evaluated their distribution by age and ethnicity, and compared the prevalence with regional and international data.

The research addresses an important public health gap by providing baseline HPV genotype data for a population in which national-level surveillance is still limited. However, while the topic is relevant and the dataset valuable, the manuscript requires some important revisions in methodological clarity, data presentation, and analytical interpretation to meet PLOS ONE standards of rigor and transparency.

Strengths:

(a) Relevance and Timeliness:

The study focuses on HPV epidemiology in the UAE - an area where such molecular data are scarce but urgently needed to inform vaccine and screening strategies.

(b) Technical Application:

The use of PNA-based fluorescence melting curve analysis introduces a sensitive molecular approach to HPV genotyping, offering methodological novelty in the regional context.

(c) Ethical and Regulatory Compliance:

Ethical approval and data anonymization procedures are appropriately described, meeting PLOS ONE9 requirements.

(d) Comprehensive Dataset:

The study includes a relatively diverse sample across multiple Emirates and ethnic groups, offering useful comparative data for future epidemiological assessments.

Weaknesses:

Methodological Detail and Transparency:

The methods section is overly technical and lacks a clear sample size justification. Statistical handling of confounding factors (age, ethnicity) and multiple comparisons is insufficiently explained.

Data Presentation:

Results are extensive but cumbersome, with multiple overlapping tables and inaccessible figures (hyperlinks not functioning). The text often repeats numerical data rather than synthesizing findings.

Interpretation and Discussion:

The discussion section focuses heavily on descriptive comparisons without critically analyzing why prevalence differences occur, what the observed genotype patterns imply, or how they relate to vaccination or screening policies.

Public Health Context:

There is limited discussion of the implications for HPV vaccination programs or national screening guidelines, missing an opportunity to link molecular findings to preventive health outcomes.

Detailed Comments to Authors

1. Title and Abstract

- Revise the title for brevity and clarity (e.g. “Genotypic Distribution and Molecular Epidemiology of HPV in Women in the UAE Using PNA-based RT-PCR”).

- Streamline the abstract by reducing numerical data in the results and adding brief mention of study type, sample size, and ethical approval.

- Revise the conclusion to emphasize implications for HPV screening and vaccination in the UAE.

2. Introduction

- Provide a clearer statement of the research gap and study hypothesis—specifically, the lack of UAE-specific HPV genotype data derived from molecular methods.

- Expand on the justification for using PNA-based fluorescence melting curve analysis, highlighting its advantages over conventional genotyping (e.g., increased sensitivity, multiplexing capability).

- Ensure citations are correctly formatted and up-to-date, replacing local file paths with proper references.

3. Methods

- Condense detailed descriptions of DNA extraction and RT-PCR protocols; move extended procedural steps to supplementary files.

- Include a sample size rationale or post-hoc power discussion to justify the study’s scale.

- Describe how unavailable data and potential confounders (age, ethnicity) were managed.

- Present 95% confidence intervals for proportions to enhance analytical robustness.

- Expand the statistical analysis section: specify tests used, significance thresholds, and handling of multiple comparisons.

4. Results

- Simplify presentation by focusing on summary statistics and moving highly-detailed breakdown (such as genotype stratifications) to supplementary tables.

- Interpret key findings (e.g., significance of p-values, age distribution trends) directly in text.

- Include confidence intervals alongside prevalence estimates to communicate statistical certainty.

- Highlight main findings succinctly—avoid repeating numerical results from tables verbatim.

5. Discussion

- Improve the discussion by including:

(a) Determinants of prevalence variation within UAE and in comparison to regional/international studies.

(b) Epidemiological interpretation of genotype distribution, particularly the predominance of HPV53.

(c) Public health implications, including relevance for HPV vaccine coverage and screening triage.

(d) Study limitations and strengths, with attention to retrospective design and sampling bias.

- Discuss the biological and vaccine policy relevance of HPV53, which is not covered in current vaccine formulations.

- Expand on differences observed between Arab and non-Arab populations, linking these to socio-behavioral or healthcare access factors.

- Avoid reiterating numerical data already presented in the Results section.

6. Conclusion

- Rewrite the conclusion to be concise and forward-looking.

- Focus on how this study contributes to the understanding of HPV epidemiology and informs policy and prevention efforts.

- Avoid listing prevalence percentages; instead, summarize key implications and suggest future directions such as:

(a) A UAE-wide HPV surveillance initiative.

(b) Expansion of vaccine formulations to include regionally prevalent genotypes (e.g., HPV53).

(c) Integration of molecular genotyping into national cervical screening programs.

.

Reviewer #1: **Yes:** Eyaya Misgan Asress /MD, genecology oncologist/Eyaya Misgan Asress /MD, genecology oncologist/Eyaya Misgan Asress /MD, genecology oncologist/Eyaya Misgan Asress /MD, genecology oncologist/

Reviewer #2: No

---

## [Author Response · Author response to Decision Letter 1]

23 Dec 2025

Responses to the Reviewers

Responses to the Editor's Comments

1) There is significant textual overlap between the current study and the preceding study (reference 13).

Odeh et al. (2023) Exploring the prevalence of Human Papillomavirus (HPV) genotypes in PAP smear samples of women in northern region of United Arab Emirates (UAE): HPV Direct Flow CHIP system-based pilot study. PLOS ONE 18(9):e0286889. https://doi.org/10.1371/journal.pone.0286889

Answer: The current study and the study by Odeh et al. were conducted in the same geographical region; however, they differ in time period and study population. Although both studies employed the same study design, the present study includes a larger number of provinces compared to the previous work. Importantly, the current study does not incorporate data from the earlier study population or from referral patients. In the revised manuscript, textual overlap with our previously published 2023 study (Odeh et al.) has been eliminated to ensure originality and clarity.

2) Some paragraphs are identical in both manuscripts. The presentation of several figures and tables appears to be substantially identical as well. Beyond removing the technical overlap, it is also necessary to provide some justification as to why the new (replication) study was needed as per PLOS criteria for publication subheading “Replication Studies”https://journals.plos.org/plosone/s/criteria-for-publication (same goal, same population, same methodology except the last HPV typing assay step, same layout and presentation, only slight details in the conclusion with much of the text copy pasted verbatim).

Answer: Replication studies

a) The objectives of the current study align with those of the previous study conducted by Odeh et al.; however, the study populations are distinct and were not referenced in the earlier work. The present study includes a larger sample size (n = 229) comprising participants of diverse nationalities. Individuals with a prior HPV diagnosis, referral cases, and follow-up patients were excluded from the current study to ensure a population of newly screened participants.

b) Cytological screening in the current study was performed using the SurePath liquid-based cytology system, and cytological grading was carried out according to the 2014 Bethesda System. The same approach was applied to all Pap smear samples collected using liquid-based cervical cytology and was consistent with the methodology used in the previous study to diagnose cytological abnormalities. However, HPV genotyping in the current study was conducted using a real-time PCR (RT-PCR) technique, whereas the previous study employed a reverse hybridization method.

c) In the manuscript, Table No. 1 has been shifted to the supplementary file as recommended by the editor. Table no.2 has a similar type of variables to the previous study conducted by us, but with different study populations at a different time period in the same region. Table no 3 has been added in the new format with a statistically significant association between age groups, nationality, and HPV status among participants, and with a strong There was a very strong, statistically significant association between HPV genotypes and HPV status. The above-mentioned tables and statistical analysis are given in the updated manuscripts – Table 2. Association of age groups, nationality, and genotypes with HPV status. Line no. 260.

3) P6 L 112-128. Study design/Data collection/Population sections.

i) Are the 229 samples all samples collected in 12 months from multiple hospitals?

Answer: In this study, A total of 229 PAP liquid-based cytological samples were collected from hospitals across multiple regions of the UAE, including the provinces of Ajman, Sharjah, Dubai, Umm Al Quwain, and Fujairah, for HPV screening and associated cytological evaluation during the study period from Jan 2024 to Jan 2025. All samples were processed at the Department of Pathology and Microbiology, Thumbay Laboratory, Thumbay University Hospital (TUH), Ajman, UAE.

ii) Information about inclusion and exclusion criteria is missing.

Answer: The inclusion and exclusion criteria of the study samples are updated in the manuscript, line no. 133 – 137, 147-149 & 151 - 153.

iii) What were the reasons patients visited hospitals (line 113)? Was this a screening or a referral population?

Answer: The study populations who visited the hospitals for primary screening purposes due to the following symptoms: watery discharge, bleeding, pelvic pain, and pain during sexual intercourse. Those study populations are considered, and the patients with a known history of HPV infection or cervical cancer, as well as those previously diagnosed with HPV and undergoing routine follow-up examinations, were excluded from this study.

4) P8 L168-169 the assay information (RealTyperTM HPV 32 Genotyping kit) is not consistent with reference 17 provided for the method (“detecting 40 types”). Assay version should be disclosed in the methods section in case the company makes further changes to the product sold.

Answer: The correct reference link for the assay information (RealTyperTM HPV32 Genotyping kit) has been incorporated in the reference section of the revised manuscript. Reference no: 17. Line no. 484

https://www.hlbpanagene.com/_ENG/html/dh_product/prod_view/86/?cate_no=2.

5) P8 L147 assay name doesn’t need to be repeated in the same paragraph.

Answer: In P8, L147, the assay name has been removed from the sentence, and the updated sentence is mentioned in the manuscript, line no. 167 – 169.

6) P9 results section. The study presents numerous highly overlapping tables (3) and figures (9) that often show high granularity of the data (ie, Table 1 has 80 cells of data, of which only 16 have 5 or more cases). Instead, consider including a supplementary Excel table with detailed information and focus the results section only on critical findings. Also, it is very difficult to find the overall prevalence of individual types, which are later referred to in the text.

Answer: As mentioned in the Editor's comments, Table 1 (Detection of HPV infection in different cytology grading in various age groups of study population) has been moved into the supplementary Table 3. Table 3 has been updated with chi-square and p-values for the association of age groups, nationality, and genotypes with HPV status of the study population, in the new Table 2 - Association of age groups, nationality, and genotypes with HPV status. Line no. 260. Figure 9 has been removed from the updated manuscript.

7) Percentages should be rounded to 1 decimal place in the results text and tables.

Answer: All the percentage values are rounded to 1 decimal place in the entire updated manuscript and but in unavoidable situations, some percentage values are still in one-digit format, like 4.6%.

8) P10 Table 1 has a marginally different age grouping than the previous study. Why was the grouping changed? From a statistical point, the >51 category is too small compared to other categories and could have been combined with the 41-50 one. Alternatively, age grouping of previous large-scale reference studies should be used, and the current data compared to those (ie, later mentioned Bruni et al, 2010)

Answer: In the present study, the study populations’ age group was originally categorised into 20 – 30 (n=77), 31-40 (n=96), 41-50 (n=48), and ≥51 (n=8). In the 1st age group (20 -30), almost all the study participants are above 21 years old, except 1 participant is 20 years old. Hence, the age group of the study participants is grouped in the above-mentioned pattern. As referred by the editor, Bruni et al 2010, in their study, the sample size was 27,343 under the age group of ≤25 years. But in our study, below ≤25 years old study participants was very low (n=5). So, the study has been grouped into 20 -30-year-olds.

As per the editor’s recommendation, the participants are too small under the age group of 51 and above (≥51; n=8), therefore, this age group has been merged with the 41-50 years old study group and updated as ≥41 years old study group (n=56). The updated age group is mentioned in the revised Tables (1, 2 & 3), results, and discussion section of the updated manuscript.

9) P14 L237 the figure 6 should be replaced by a table (or at least the summary information provided as a supplement table)to make it more readable.

Answer: Figure 6 has been replaced as Table 1. Frequency of high and low-risk genotypes in HPV positive samples, in the manuscript, Line no. 232.

10) P17 L302-306 the p value was calculated on the total number of positive and negative cases. This is not shown on Table 3, where the same p-value details are shown.

Answer: Table 3 has been updated to Table 2, which includes additional statistical outputs from the chi-square test, including corresponding p-values, to assess the association between the dependent variable (HPV status) and the independent variables (age groups, nationality, and genotypes). This revised table is now incorporated into the updated manuscript, and the statistically significant associations between HPV status and the independent variables are clearly reported in the Results section. Line no. 262 – 272.

11) P18 Discussion. Critically, unless the type of population studied (general screening, referral, convenience) is known, any comparisons with previous studies is impossible to appreciate or understand.

Answer: The current study, while comparing with the previous study conducted by our team (Odeh et al., Ref. 13) reported several HPV genotypes were reported in combined forms, such as 62/81, 6/43, 31/68, 44/55, and 52/86. These combination results were due to limitations of the reverse hybridization diagnostic method, which did not allow for precise differentiation of individual genotypes. In contrast, the current study accurately identified each genotype independently using an enhanced diagnostic approach – RT-PCR, confirming individual genotypes such as LR81, 6, 43, and 44, as well as HR31, 68, and 52. It is mentioned in the revised manuscript – line no. 356 – 361.

12) The prevalence data are not put in the context of the world in general. For example, Bruni L et al, Cervical Human Papillomavirus Prevalence in 5 Continents: Meta-Analysis of 1 Million Women with Normal Cytological Findings, The Journal of Infectious Diseases, Volume 202, Issue 12, 2010, Pages 1789–1799, https://doi.org/10.1086/657321. or a similar encompassing study.

Answer: As mentioned by the editor, similar encompassing data of the most prevalent of HPV genotypes has been categorised region-wise: MENA, Asia, and Central Asia, and given in Table 3 (Comparative frequency of the most prevalent HPV Genotypes among HPV-positive women with ethnicity and nationality). Line no. 276

The most prevalent HPV genotypes of different geographical regions are also discussed in the discussion section and updated in the revised manuscript. Line no. 314 – 324 & 333 - 355.

13) The most commonly found HPV53 was not put into sufficient context in the discussion or conclusion.

Answer: The most commonly encountered genotype HR53 has been discussed with previous studies conducted in the UAE and globally with Sanjos et al (2007), Burni et al (2010), Osmani et al 2025 and a study from China (2023). These discussion parts have been updated in the revised manuscript Line no. 319 – 324.

14) The discussion doesn’t sufficiently address the differences between Arab and non-Arab participants and thus doesn’t clearly state whether any differences are notable or if such differentiation is irrelevant for future studies

Answer: The above-mentioned comments are considered, and the difference between Arab and non-Arab participants was clearly mentioned and revised in the discussion section. Line no. 333 – 355.

15) Supplementary file S2 doesn’t answer the question of potentially eligible participants. The file completely overlaps the manuscript text and can be removed as uninformative or revised.

Answer: The supplementary file S2 has been revised, and the revised file S2 has been submitted along with the updated manuscript.

Responses to Reviewer-1 comments

1) The authors tried to highlight the geographic distribution of HPV genotypes among PAP smear samples by using the melted PCR technology. Their finding underscores the importance of molecular testing for HPV genotyping. One comment to highlight is the distinction of the samples as Arab and non-Arab is a crude data i.e. it won't give a casual inference, as the details of these populations is not known, including the epidemiological profile, sexual history etc.

Answer: We appreciate the reviewer’s insightful comment. The current study primarily focused on the initial HPV screening of the study population, categorized into Arab and non-Arab ethnic groups. As this was a retrospective, laboratory-based investigation, detailed epidemiological profiles and sexual history information could not be retrieved from the available medical records. This limitation has now been clearly acknowledged and updated in the revised manuscript. Line no. 410 – 421.

We agree that such data are important and will ensure that future studies incorporate comprehensive epidemiological and sexual history information, obtained with proper informed consent from the study participants.

Response to Reviewer – 2 comments

2) Methodological details and transparency

i. The methods section is overly technical and lacks a clear sample size justification.

Answer: The sample size of the study population has been updated and clearly mentioned in the subheading – “Study populations” under the Methods section of the updated manuscript. Line No: 143 - 158.

ii) Statistical handling of confounding factors (age, ethnicity) and multiple comparisons is insufficiently explained.

Answer: The confounding factors, such as Age and ethnicity of the study population, have been updated in tables 2 & 3 of the revised manuscript.

Table 2. Association of age group, nationality, and genotypes with HPV status. Line no. 260

Table 3. Comparative frequency of the most prevalent HPV genotypes among HPV-positive women with ethnicity and nationality. Line no. 276.

3) Data Presentation

i) Results are extensive but cumbersome, with multiple overlapping tables and inaccessible figures (hyperlinks not functioning).

Answer: In accordance with the recommendations, Tables 1 & 2 have been moved to the supplementary file, and Figures 6 and 9 have been removed from the revised manuscript. The content originally presented in Figure 6 has now been reformatted and included as Table 1. “Frequency of high and low-risk genotypes in HPV positive samples,” in the updated version. Line no. 232.

All required figures have been prepared in TIF format and submitted as separate files, following the PLOS ONE submission standards. Figure captions remain clearly presented within the main manuscript, while the corresponding figures are provided as separate attachments in the updated submission package. Now all the hyperlinks are functioning.

ii) The text often repeats numerical data rather than synthesizing findings.

Answer: In response to the comment regarding repeated numerical data, we have revised the manuscript to minimize redundancy. Essential numerical findings are now consolidated and presented clearly in Tables 1, 2, and 3 within the main text, while additional supporting tables have been moved to the supplementary file. This restructuring ensures better synthesis of the results and improves the overall readability of the revised manuscript.

4) Interpretation and discussion

The discussion section focuses heavily on descriptive comparisons without critically analyzing why prevalence differences occur, what the observed genotype patterns imply, or how they relate to vaccination or screening policies.

Answer: In the revised manuscript, we have clarified the demographic context of the United Arab Emirates, where Emirati nationals constitute approximately 15% of the population, while the remaining 85% comprises expatriates of diverse nationalities, predominantly from Asian countries.

Our analysis revealed t

---

## [Decision Letter · Decision Letter 1]

28 Jan 2026

Dear Dr. Rahamathullah,

We look forward to receiving your revised manuscript.

Kind regards,

Ivan Sabol

Academic Editor

PLOS One

Journal Requirements:

**Additional Editor Comments:**

The revised manuscript addresses many of the issues and the authors efforts are appreciated and must be acknowledged. Detailed methodological descriptions are helpful. Addition of statistics expert and regression is appreciated

Some issues remain or were inadvertently introduced in the revision process but are mostly of technical nature.

1. The combined PDF file apparently contains both original as well as revised documents (and material). Old documents should be removed so that it is clear what is the final document version and which images are finally included. Currently there are still 9 figures even though the revised manuscript includes less. Erroneous items might lead to errors in the final steps

2. Table order was not revised fully so the first table referenced is table 2 (line 201) followed by table 4 at line 203) while table 1 has no inline reference and table 3 appears at line 276.

The table order should be revised to accommodate changes to regular and supplementary tables made in the revision.

3. The authors rounded numbers to integers (ie “8”) instead of allowing one decimal place as expected (ie “8.1”). Also Table 3 appears to have “03” format instead of integers

4. Revised Table 1 (page 11) presents percentages of each type across all HPV types, while usually when presenting “HPV prevalence” the percentage of positive cases is shown (number of samples positive divided by the number of samples tested).

Saying that HPV53 represented 8% of all identified types (16/191) is less informative and possibly misleading to clinical practitioners than saying HPV53 was found in 6.9% (16/229) screened women. The same issue should be clarified throughout the manuscript

5. Table 2 (page 13) last rows examine “HPV status” versus “HPV genotype” which doesn’t add information since groups are selected to be different (genotypes can only be in positive cases) and thus statistical comparison is unnecessary. Furthermore, it is impossible for 3 samples to be at once “negative for HPV” and within HPV status “Positive” column suggesting that some data issues remain. Instead of Genotype rows a comparison across cytological diagnosis would be more informative or this section of table 2 should be removed

6. Revised Table 3 (page 14) should include the total number of women it refers to in the table title to allow interpretation as well as totals of cohorts so that the table is standalone and doesnt require referencing Figure 7 or multiple places in text. Table 3 shows no percentages to allow interpretation. Also table makes a distinction between asia and central asia cohorts that cannot be interpreted due to very small sample numbers (and should thus remain grouped from a statistical as well as interpretation contexts). Especially since “Central asia” is not shown separately in figures (ie fig 6) or supplementary tables. The discussion does explicitly mention Thailand (ie at page 18 line 367 but “substantial number of participants from Thailand” is not shown in materials or tables where such data is expected (it is shown only on figure 7 which is very difficult to summarize). It is also very difficult to link Table 3 with information on Figure 7 since table 3 shows prevalence within positives while Figure 7 shows cases so subtotals dont match.

7. There are still some minor inconsistencies in the data. Page 12 test at lines 238-239 states that there were 39 HPV positive Arab cases and 57 non-Arabs (39+57=96). The same subtotal is within Figure 4 (49+26+18+1+2=96). However, Table 2 states there were 95 positive cases (Arab 39+ non-Arab 56=95, 95 also for age group and genotype totals)

Reviewers' comments:

Reviewer's Responses to Questions

**Comments to the Author**

Reviewer #1: All comments have been addressed

2. Is the manuscript technically sound, and do the data support the conclusions?

Reviewer #1: Yes

3. Has the statistical analysis been performed appropriately and rigorously?

Reviewer #1: Yes

4. Have the authors made all data underlying the findings in their manuscript fully available?

Reviewer #1: Yes

5. Is the manuscript presented in an intelligible fashion and written in standard English?

Reviewer #1: Yes

Reviewer #1: The authors have now significantly improved the manuscript with appropriate answers for the comments given.

.

Reviewer #1: No

---

## [Author Response · Author response to Decision Letter 2]

11 Mar 2026

Responses to the Editor's Comments

1) The combined PDF file apparently contains both original as well as revised documents (and material). Old documents should be removed so that it is clear what the final document version is and which images are finally included. Currently there are still 9 figures even though the revised manuscript includes less. Erroneous items might lead to errors in the final steps.

Author’s response: Thank you for pointing this out. In accordance with the editor’s suggestion. The previously uploaded old documents have now been removed from the author submission system to avoid any confusion. Only the updated revised manuscript (in Word format), along with the relevant figures, supplementary tables, and the author’s rebuttal letter containing detailed responses to the Editor’s comments, have been uploaded to the submission site. This ensures that only the final and correct version of the manuscript and the corresponding materials are available for review and further processing.

2)Table order was not revised fully so the first table referenced is table 2 (line 201) followed by table 4 at line 203) while table 1 has no inline reference and table 3 appears at line 276. The table order should be revised to accommodate changes to regular and supplementary tables made in the revision.

Author’s response: Thank you for highlighting this issue. The order of the tables has been carefully reviewed and revised in accordance with the Editor’s comments to ensure proper sequence and consistency with their corresponding in-text citations. In the updated manuscript, Table 2 is now correctly referenced in line 199, where a Chi-square test was conducted to determine the association between the dependent variable (HPV status) and the independent variables, including age group, nationality, and genotypes, with a p-value of <0.05 considered statistically significant. Furthermore, Table 4 is appropriately cited in line 202, where binary logistic regression analysis was performed to determine the predictors of positive HPV status, including age groups and nationalities. These revisions ensure that all tables are cited in the correct order and correspond accurately to the content presented in the manuscript.

3) The authors rounded numbers to integers (i.e., “8”) instead of allowing one decimal place as expected (i.e., “8.1”). Also, Table 3 appears to have “03” format instead of integers.

Author’s response: Thank you for bringing this to our attention. The numerical values in the tables have been carefully reviewed and corrected. In Tables 1 and 3, numbers have been appropriately formatted as integers where required. Specifically, in Table 3, values previously presented in formats such as “08” have been corrected to “8.” Similar formatting corrections have been applied consistently throughout the table to ensure clarity and uniformity in numerical presentation.

4) Revised Table 1 (page 11) presents percentages of each type across all HPV types, while, usually, when presenting “HPV prevalence,” the percentage of positive cases is shown (number of samples positive divided by the number of samples tested).

Saying that HPV53 represented 8% of all identified types (16/191) is less informative and possibly misleading to clinical practitioners than saying HPV53 was found in 6.9% (16/229) of screened women. The same issue should be clarified throughout the manuscript.

Author’s response: Thank you for this valuable comment. In accordance with the editor’s suggestion, Table 1 (page 12) has been revised to present the prevalence of each HPV genotype based on the total number of women screened (n = 229) rather than the proportion among the total detected genotypes (n = 191). This approach better reflects the actual prevalence of each HPV type in the study population.

For example, HPV53 was detected in 16 samples, corresponding to a prevalence of 6.9% (16/229) among the screened women, which has been rounded to 7% in the table.

Similarly, the percentages for all other detected HPV genotypes have been recalculated using the same denominator and updated accordingly in Table 1. The manuscript has also been reviewed to ensure that this method of reporting HPV prevalence is applied consistently throughout the text.

5) Table 2 (page 13) last rows examine “HPV status” versus “HPV genotype” which doesn’t add information since groups are selected to be different (genotypes can only be in positive cases) and thus statistical comparison is unnecessary. Furthermore, it is impossible for 3 samples to be at once “negative for HPV” and within the HPV status “Positive” column, suggesting that some data issues remain. Instead of Genotype rows, a comparison across cytological diagnosis would be more informative or this section of Table 2 should be removed.

Author’s response: In accordance with the editor’s suggestion, Table 2 (page 14) has been revised to improve clarity and presentation. The previously included row listing genotype categories—such as HPV negative, HR single, LR single, HR multiple, LR multiple, and mixed HR & LR genotypes—has been removed. The revised table now focuses only on the key variables, namely age and nationality.

Age has been categorized into three groups, and their corresponding HPV status is presented accordingly. In addition, the table now includes the results of the Chi-square analysis and the corresponding p-values to demonstrate the statistical association between the variables and HPV status. This revision ensures a clearer and more concise presentation of the data.

6) a) Revised Table 3 (page 14) should include the total number of women it refers to in the table title to allow interpretation, as well as totals of cohorts, so that the table is standalone and doesn't require referencing Figure 7 or multiple places in text.

Author’s response: Thank you for the valuable suggestion. In accordance with the editor’s suggestion, the total number of the study population (n = 229) has been added in Table 3 (page 14) as a separate row, placed above the row indicating the number of HPV-positive cases (n = 96; 42%). This addition allows the table to clearly present the study cohort and facilitates easier interpretation without requiring reference to other figures or sections of the manuscript.

Furthermore, the title of Table 3 has been revised to “Total number of the most prevalent HPV genotypes among HPV-positive women and their ethnicity.” These revisions ensure that the table is more informative and can be interpreted as a standalone element within the manuscript.

b) Table 3 shows no percentages to allow interpretation. Also, the table makes a distinction between Asia and Central Asia cohorts that cannot be interpreted due to very small sample numbers (and should thus remain grouped from a statistical as well as interpretation contexts). Especially since “Central Asia” is not shown separately in figures (i.e., fig 6) or supplementary tables.

Author’s response: Table 3 has been revised to include the prevalence percentages of both high-risk (HR) and low-risk (LR) HPV genotypes within the study cohorts. The study population is now categorized into Arab and non-Arab study groups in the updated table. Under these two categories, the most prevalent HPV genotypes are presented along with their corresponding prevalence percentages.

The percentages were calculated using the total number of screened women (n = 229) as the denominator. For example, HPV53 was detected in 11 women in the Arab cohort, corresponding to a prevalence of 5% (11/229), and in 5 women in the non-Arab cohort, corresponding to a prevalence of 2% (5/229).

Similarly, the prevalence percentages for all other detected HPV genotypes have been recalculated using the same denominator and updated accordingly in Table 3. This revision improves the clarity and interpretability of the table while maintaining consistency with the presentation of data throughout the manuscript.

c) The discussion does explicitly mention Thailand (i.e., at page 18, line 367 but “substantial number of participants from Thailand” is not shown in materials or tables where such data is expected (it is shown only on figure 7, which is very difficult to summarize). It is also very difficult to link Table 3 with information on Figure 7 since Table 3 shows prevalence within positives, while Figure 7 shows cases, so subtotals don't match.

Author’s response: Thank you for the helpful comment. In response to the editor’s suggestion, the Discussion section has been revised, particularly on page 18 (lines 363–368), to improve clarity and ensure consistency with the data presented in the tables and figures. The revised text now clearly describes the distribution of the most prevalent HPV genotypes among the study cohorts.

The updated section (page 18, lines 363–368) now reads as follows:

“Our analysis demonstrated that specific HPV genotypes, particularly HR53 (11; 5%) and HR16 (8; 3%), as well as LR61 (7; 3%), were more commonly detected among the Arab study group originating from MENA-region countries residing in the UAE. In contrast, genotypes HR68 (10; 4%), HR66 (7; 3%), HR35 (6; 3%), and LR81 (5; 2%) were more frequently observed among the Asian study group. This Asian cohort included a substantial number of participants from Thailand, where genotypes HR68 (3%), HR66 (3%), HR35 (2%), and LR81 (2%) were detected.”

Furthermore, Figure 7 illustrates the nationality-wise HPV status of the study population, presenting the number of HPV-positive and HPV-negative cases among participants from different countries. This figure provides a visual representation of the distribution of HPV status across nationalities within the study population.

7) There are still some minor inconsistencies in the data. Page 12 test at lines 238-239 states that there were 39 HPV-positive Arab cases and 57 non-Arabs (39+57=96). The same subtotal is within Figure 4 (49+26+18+1+2=96). However, Table 2 states there were 95 positive cases (Arab 39+ non-Arab 56=95, 95 also for age group and genotype totals)

Author’s response: Thank you for your valuable comment. The minor numerical inconsistencies identified in the manuscript have been carefully reviewed and corrected. The relevant sections of the manuscript and the tables have been updated to ensure consistency in the reported data.

Specifically, the text on page 11 (lines 235–238) has been revised to correctly state the distribution of participants and HPV status as follows:

“Overall, 114 (50%) participants were Arabs and 115 (50%) were non-Arabs. HPV positivity was observed in 39 Arabs (17%) and 57 non-Arabs (25%), while 75 Arabs (33%) and 58 non-Arabs (25%) tested HPV-negative.”

In addition, Table 2 previously contained minor numerical discrepancies in the distribution of HPV-positive and HPV-negative cases across different age groups and nationality categories. These values have now been carefully corrected and updated in the revised Table 2 to ensure that the totals are accurate and consistent with the data presented in the manuscript text and figures.

---

## [Editor Report · Decision Letter 2]

15 Mar 2026

Genotypic Distribution and Molecular Epidemiology of HPV in Women in the UAE using PNA-based RT PCR"

PONE-D-25-39382R2

Dear Dr. Rahamathullah,

We’re pleased to inform you that your manuscript has been judged scientifically suitable for publication and will be formally accepted for publication once it meets all outstanding technical requirements.

Kind regards,

Ivan Sabol

Academic Editor

PLOS One
---

## [Editor Report · Acceptance letter]

PONE-D-25-39382R2

PLOS One

Dear Dr. Rahamathullah,

I'm pleased to inform you that your manuscript has been deemed suitable for publication in PLOS One. Congratulations! Your manuscript is now being handed over to our production team.

Kind regards,

on behalf of

Dr. Ivan Sabol

Academic Editor

PLOS One